# Effects of Restraint Stress on Circulating Corticosterone and Met Enkephalin in Chickens: Induction of Shifts in Insulin Secretion and Carbohydrate Metabolism

**DOI:** 10.3390/ani14050752

**Published:** 2024-02-28

**Authors:** Colin G. Scanes, Krystyna Pierzchała-Koziec, Alina Gajewska

**Affiliations:** 1Department of Biological Science, University of Wisconsin Milwaukee, Milwaukee, WI 53211, USA; 2Department of Animal Physiology and Endocrinology, University of Agriculture, Mickiewicza 24/28, 30-059 Kraków, Poland; 3Kielanowski Institute of Animal Physiology and Nutrition, Polish Academy of Sciences, 05-110 Jablonna, Poland; a.gajewska@ifzz.pl

**Keywords:** stress, insulin, glucose, corticosterone, Met-enkephalin, insulin resistance

## Abstract

**Simple Summary:**

Chickens are sensitive to different stressors such as emotional stressors, feed reduction or restraint. These may influence physiological parameters including those related to metabolism, growth and development. The choice of reliable markers of stress is essential to evaluate poultry welfare. Therefore, the objective of this study was to examine whether measuring glucose and insulin levels could be useful for establishing the short-term effects of stress. Additionally, we measured the changes in Met-enkephalin, an endogenous opioid, and corticosterone, hormones responsible for the stress response. Our results show that restraint is a severe stressor stimulating both the hypothalamo–pituitary–adrenocortical axis and the opioid system activity, together with influencing carbohydrate metabolism.

**Abstract:**

This study examined the effects of acute restraint stress in the presence or absence of naltrexone on the circulating concentrations of insulin, glucose, Met-enkephalin and corticosterone in 14-week-old chickens [design: 2 sex × 2 stress/non-stress × 2 +/− naltrexone]. In chickens (five male and five females per treatment) subjected to restraint for 30 min, there were increases in the plasma concentrations of corticosterone and Met-enkephalin. The plasma concentrations of insulin and glucose were also increased in the chickens during restraint. Moreover, there were increases in the plasma concentrations of insulin and glucose in the chickens. The patterns of expression of the proenkephalin gene (PENK) in both the anterior pituitary gland and the adrenal gland were very similar to that of plasma Met-enkephalin. There were relationships between the plasma concentrations of corticosterone, Met-enkephalin, insulin and glucose after 30 min of restraint. The effects of naltrexone treatment on both untreated and stressed chickens were also examined, with naltrexone attenuating the stress-induced increases in the plasma concentrations of corticosterone, Met-enkephalin and glucose but not in those of insulin. The present study demonstrates that stress increases insulin secretion in chickens but also induces insulin resistance.

## 1. Introduction

Restraint stress is a widely used stressor in animals including rodents [1,2], reviewed in [3], and chickens [4,5,6,7,8,9,10,11]. This stress does not involve physical pain but represents a psychologically aversive inescapable environment, also restricting movement [12]. Another aspect of restraint stress is the exposure to a novel environment, such as small individual cages [12]. Assays for anxiety- and depression-related behaviors have been developed [13,14]. Anxiety-related behaviors in mice can be assessed by evaluating their avoidance of heights and open spaces and their preference for dark enclosed spaces in an elevated plus maze [13,14], together with hyperactivity, for instance, in light/dark transition tests and open field tests [14,15]. In mice, depression-related behaviors are studied using the Porsolt forced swim test, the tail suspension test and the sucrose preference test [14,15]. Repeated exposure of rodents to restraint stress is followed by reduced anxiety-like behavior and increased depression-like behavior [14,15]. Restraint stress also results in anorexia and weight loss in rodents [16,17]. Corticotropin-releasing hormone (CRH) plays a central role in the response to restraint stress, activating the hypothalamo–pituitary–adrenocortical (HPA) axis, thereby increasing the circulating concentrations of glucocorticoids [15,18,19].

Poultry are sensitive to multiple psychological stressors, reviewed in [20], with the circulating concentrations of corticosterone having been reported to be increased by restraint [4,5,6,7,8,9,10,11], withdrawal of feed [21,22] and exposure to *Escherichia coli* lipopolysaccharide [23,24] in chickens, as well as herding in turkeys [25]. The difficulty of attempting to transpose stress responses across species is exemplified by the different responses to shackling in chickens and turkeys, with marked increases in feather flapping and the circulating concentrations of corticosterone in chickens [26,27] but not in turkeys [24,27]. The activation of the HPA in chickens decreases growth [28,29] and enhances adiposity [30] and increases triglyceride lipase in the adipose tissue [22]. These effects are undesirable in poultry production. It is important to develop and employ optimal markers of stress to evaluate the welfare of chickens. What is not known is whether restraint stress influences the plasma concentrations of Met-enkephalin, insulin and glucose. The present study examines the effects of restraint stress on the plasma concentrations of Met-enkephalin and corticosterone, the latter determined to confirm a stress effect. Effects on the expression of the proenkephalin gene (PENK) in both the anterior pituitary gland and the adrenal gland were also determined.

There is considerable evidence that exogenous glucocorticoids induce both insulin secretion and insulin resistance in chickens. The circulating concentrations of insulin are elevated in chickens receiving corticosterone administration [31,32,33,34]. Similarly, dexamethasone increased the plasma concentration of insulin in chickens refed for 3 h after fasting for 24 h but not in fasted birds [35]. It is possible that the effects of exogenous glucocorticoids represent pharmacological (i.e., extremely high doses that are outside the physiological range) rather than physiological effects. What is not known is whether stressors influence insulin secretion and/or insulin resistance in chickens. The present study also examines the effects of the opioid antagonist naltrexone on the stress response (plasma concentrations of corticosterone and Met-enkephalin) and on the circulating concentrations of glucose and insulin in both stressed and control chickens.

## 2. Materials and Methods

All animal procedures were conducted with prior institutional ethical approval in accordance with the Local Institutional Animal Care and Use Committee (IACUC). The animal study protocol 120/2013 was approved by the Institutional Review Board and the First Local Ethical Committee on Animal Testing in Krakow, Poland. 

### 2.1. Animal Model

The study employed twenty female and twenty male 14-week-old ISA Brown hybrid (Rhode Island x Leghorn) egg-laying chickens weighing 1.2 ± 0.10 kg. The birds were maintained in individual cages (60 × 60 × 60 cm) in a controlled environment (photoperiod 12L/12D with lights on from 7 a.m. to 7 p.m.) and at room temperature (20 °C). The chickens received feed and water ad libitum. The animals were habituated to these conditions for 7 days before the experiments. The chickens were randomly assigned to the experimental groups [design: 2 (sex) × 2 (stress/non-stress) × 2 (+/− naltrexone)]. The treatment groups (*n* = 5 for each sex per treatment) were subjected to the following conditions: control (O), stress by restraint (Re), naltrexone treatment (Na), and naltrexone treatment plus restraint (Re + Na).

### 2.2. Experimental Design

The overall design of the study was the following: 2 (sex) × 2 (stress/non-stress) × 2 (+/− naltrexone). Pretreatment blood samples were taken, and then the chickens received an i.v. injection of either 0.9% saline (control O and group Re to be stressed) or naltrexone (2 mg/kg b.w., Sigma-Aldrich, St. Louis, MO, USA; treatment groups Na and Re +Na). The chickens were separately kept in boxes (30 × 30 × 30 cm) with no light and sound for 30 min. To collect the blood samples, the birds were transitorily manually held, and the samples (each of 2 mL) were taken from the left wing-vein and transferred into heparinized tubes at the following times: 15 min before the initiation of the stress or sham treatment, 30 min after the initiation of restraint and 10 min after terminating the stress. The blood samples were taken at the same times starting at 9 a.m. Plasma was obtained after centrifugation (3000 rpm, at 4 °C, for 20 min) and immediately frozen at −80 °C until further processing.

### 2.3. Hormone and Glucose Assays

The concentrations of Met-enkephalin in the plasma were determined by radioimmunoassay employing the method of Pierzchala-Koziec and colleagues [36,37]. The inter-assay and intra-assay coefficients of variance were, respectively, 7% and 11%. The circulating levels of corticosterone were measured in duplicate by a radioimmunoassay (RIA) using the Corticosterone Double Antibody RIA kit (07–120102, MP Biomedicals, Irvine, CA, USA) in 10 µL of plasma. The intra-assay and inter-assay coefficients of variance were 9.1% and 15.5%, respectively. The plasma levels of insulin were estimated by a radioimmunoassay using a commercial kit obtained from DIAsource ImmunoAssays S.A., Ottignies-Louvain-la-Neuve, Belgium (INS-IRMA KIP-125I). The intra-assay and inter-assay coefficients of variance were 6.6% and 14.4%, respectively. The glucose concentrations in plasma were determined using an enzymatic colorimetric method (Glucose kit; Sigma Diagnostics, Livonia, MI, USA). The intra-assay and inter-assay coefficients of variation were less than 4.0%.

### 2.4. PENK (Met-Enkephalin Gene) Expression

mRNA gene expression was estimated by a modification of a reported method [38]. Briefly, the frozen fragments of the pituitary and adrenal glands were sliced (14 µm sections) using a Leica cryostat microtome (−22 °C). The sections were thaw-mounted on gelatin-covered microscopic slides and stored for 3 days at −20 °C before the assay. Then, the tissue sections were thawed and fixed in 4% formaldehyde in phosphate-buffered saline (PBS; pH 7.4) for 10 min. The sections were acylated for 10 min in triethanolamine/acetic anhydride (0.25%). The sections were dehydrated by immersion in graded ethanol (70%, 80%, 95%, 100%) and air-dried. After pre-hybridization, a synthetic deoxyoligonucleotide, complementary to a fragment of the rat proenkephalin (PENK) gene, was labeled using 35S-dATP (1200 Ci/nmol) to obtain a specific activity of about 4 × 10^6^ cpm μL^−1^. The probes were diluted in hybridization buffer (formamide, dextran sulfate, saline–sodium citrate buffer (SSC), Denhardt’s solution, yeast tRNA, herring sperm DNA). Hybridization occurred during 20 h in a humidified chamber at 37 °C. Then, the sections were washed once in SSC for 10 min and four times for 15 min each in SSC/50% formamide at 40 °C and then were rinsed in SSC and distilled water at room temperature and air-dried. The sections were exposed to a Kodak film for four weeks (−80 °C). The photo-stimulated luminescence (PSL) density of the irradiated plates was measured with a BAS-1000 readout system. The PSL/mm^2^ in the resultant film images was determined using a computer image analysis system.

### 2.5. Statistical Analysis

The data were checked for normal distribution (DATAtab, https://datatab.net/statistics-calculator/hypothesis-test, accessed 7 February 2024). The data obtained at different times from the same birds were analyzed by repeated-measures ANOVA. The data at the 30 min (following 30 min of restraint or sham treatment) sampling time were analyzed by three-way ANOVA (DATAtab https://datatab.net/statistics-calculator/hypothesis-test, accessed 7 February 2024) followed by the Tukey’s honestly significant difference test as the range test. Where appropriate (concentrations of hormones or glucose versus time or in pairwise comparisons between concentrations of hormones or glucose), the data were analyzed by linear regression. 

## 3. Results

### 3.1. Basal (Pre-Treatment) Concentrations of Corticosterone, Met-Enkephalin, Glucose and Insulin

The basal plasma concentrations of Met-enkephalin were 12.9% greater (*p* < 0.0001) in male than in female chickens (Table 1). Similarly, the plasma concentrations of glucose were 12.8% greater (*p* < 0.0001) in male than in female chickens (Table 1). However, there were no differences between the plasma concentrations of either corticosterone or insulin between female and male chickens (Table 1). 

There were no relationships (*p* > 0.05) between the basal (pre-treatment) plasma concentrations of corticosterone, Met-enkephalin, insulin and glucose, except for the plasma concentrations of Met-enkephalin and those of glucose in male or female chickens.

### 3.2. Effects of Restraint Stress and/or Naltrexone Administration 

Table 2 summarizes the effects of restraint stress and/or naltrexone administration on the plasma concentrations of corticosterone, Met-enkephalin, insulin and glucose in 14-week-old chickens. The plasma concentrations of corticosterone were elevated (*p* < 0.001) compared to the pre-treatment ones after 30 min of restraint stress in juvenile chickens (Table 2; Figure 1), the increases being of 82.9% in females and 100.0% in males. Moreover, the plasma concentrations of corticosterone were increased (*p* < 0.001) compared to the pre-treatment ones in chickens receiving the opioid antagonist naltrexone (Figure 1), the increases being of 67.1% in females and 63.0% in males. In the presence of naltrexone, the increases compared to the pre-treatment values in the plasma concentrations of corticosterone were attenuated in both female and male chickens (Table 2; Figure 1), the decreases relative to restraint alone being of 29.7% in females and 29.0% in males. The increases in the plasma concentrations of corticosterone compared to the pre-treatment values were greater (*p* < 0.001) in males than in female chickens. 

The plasma concentrations of Met-enkephalin were increased (*p* < 0.001) compared to the pre-treatment ones after 30 min of restraint in chickens (Table 2; Figure 1), by 76.7% in females and by 120.1% in males. The plasma concentrations of Met-enkephalin were greater (*p* < 0.001) in stressed males than in stressed females (Table 2; Figure 1). In the presence of naltrexone, there were reduced (*p* < 0.001) responses to restraint in the plasma concentrations of Met-enkephalin (Table 2; Figure 1), the decreases being of 50.6% in females and 32.6% in males.

The plasma concentrations of insulin were markedly elevated (*p* < 0.001) compared to the pre-treatment ones in chickens subjected to restraint for 30 min (Table 2; Figure 2), the increases being of 20.3-fold in females and 14.2-fold in males. The magnitude of the increases in the plasma concentrations of insulin compared to the pre-treatment levels was greater (*p* < 0.001) in females than in males (Table 2; Figure 2). The plasma concentrations of glucose were elevated (*p* < 0.001) compared to the pre-treatment ones in both female and male chickens after 30 min of restraint (Table 2; Figure 2), the increases being of 64.0 % in female and 115.5 % in male chickens. The plasma concentrations of glucose in stressed birds were greater (*p* < 0.001) in males than in females (Table 2; Figure 2). The plasma concentrations of glucose were increased (*p* < 0.001) compared to the pre-treatment values in male but not in female birds receiving the naltrexone treatment (Table 2; Figure 2). In contrast, there was a smaller (*p* < 0.001) increase in the plasma concentrations of glucose in birds subjected to restraint when receiving naltrexone (Figure 2).

### 3.3. Effects of Recovery following Restraint Stress

The plasma concentrations of both corticosterone and Met-enkephalin declined (*p* < 0.001) during the 10 min recovery period compared to those during restraint stress (Table 2). The magnitudes of the reductions were greater (*p* < 0.001) for the plasma concentrations of corticosterone than for those of Met-enkephalin and in female than in male chickens (Table 2), the decreases being of 49.2% in female and 27.4% in male chickens for the plasma concentrations of corticosterone and of 27.4% and 8.7% for the plasma concentrations of Met-enkephalin in female and male chickens (Table 2). Similarly, the plasma concentrations of glucose in birds that had been subjected to restraint declined (*p* < 0.001) during the recovery period compared to those during restraint stress, the decreases being of 23.7% in females and 45.0% in males (Table 2). In contrast, the plasma concentrations of insulin were unchanged during the short recovery period (Table 2).

### 3.4. Effect of Repeated Blood Sample Collection in the Controls and Following Naltrexone Administration

In both control female and male chickens that had been subjected to multiple blood sampling, there were increases in the plasma concentrations of glucose and insulin with time but not in those of corticosterone or met-enkephalin. For instance, the plasma concentrations of glucose were increased by 11.3% in females and 6.7% in males compared to the pre-treatment ones for up to 40 min after sample collection [females: adjusted R^2^ = 0.833 (*p* < 0.001), males: adjusted R^2^ = 0.793 (*p* < 0.001)]; increases were also observed in the plasma concentrations of insulin [females: adjusted R^2^ = 0.396 (*p* < 0.05), males: adjusted R^2^ = 0.621 (*p* < 0.001)]. 

### 3.5. PENK Expression 

There was increased (*p* < 0.05) PENK expression in both anterior pituitary and adrenal tissue in both male and female chickens subjected to restraint (Figure 3). In contrast, in both male and female chickens, PENK expression was reduced (*p* < 0.05) after naltrexone administration (Figure 3). There were increases (*p* < 0.05) in PENK expression in both adrenal and anterior pituitary tissue in chickens subjected to restraint and receiving naltrexone administration compared to chickens receiving naltrexone alone (Figure 3). 

### 3.6. Relationships between the Plasma Concentrations of Corticosterone, Met-Enkephalin, Insulin and Glucose Together with PENK Expression

There were close relationships between the plasma concentrations of corticosterone and those of other hormones and glucose, as well as between those of different hormones or the same hormone and between those of hormones and glucose. For instance, there were relationships at the 30 min time point between the plasma concentrations of corticosterone and those of Met-enkephalin [adjusted R^2^ = 0.867 (*p* < 0.001)], those of insulin [adjusted R^2^ = 0.499 (*p* < 0.001)] and of glucose [adjusted R^2^ = 0.118 (*p* < 0.05)], between the plasma concentrations of Met-enkephalin and those of insulin [adjusted R^2^ = 0.307 (*p* < 0.001)] and glucose [adjusted R^2^ = 0.835 (*p* < 0.001)] and between the plasma concentrations of insulin [adjusted R^2^ = 0.453 (*p* < 0.001)].

There were close relationships between the expression levels of PENK in the adrenal and anterior pituitary glands [adjusted R^2^ = 0.867 (*p* < 0.001]) and between the PENK expression levels in the adrenal and anterior pituitary glands and the plasma concentrations of corticosterone (at 30 min: respectively, adjusted R^2^ = 0.797 (*p* < 0.001) and = 0.920 (*p* < 0.001); at 40 min: respectively, adjusted R^2^ = adjusted R^2^ = 0.613 (*p* < 0.001) and = 0.757 (*p* < 0.001)), of Met-enkephalin (at 30 min: respectively, adjusted R^2^ = 0.921 (*p* < 0.001) and 0.758 (*p* < 0.001); at 40 min: adjusted R^2^ = 0.736 (*p* < 0.001) and 0.686 (*p* < 0.001)), of insulin at 30 min: adjusted R^2^ = 0.138 (*p* < 0.05) and 0.216 (*p* < 0.001); at 40 min: adjusted R^2^ = 0.101 (*p* < 0.1) and 0.198 (*p* < 0.001)) and of glucose (at 30 min: adjusted R^2^ = 0.508 (*p* < 0.001) and 0.668 (*p* < 0.001); at 40 min: adjusted R^2^ = 0.482 (*p* < 0.001) and 0.630 (*p* < 0.001)).

## 4. Discussion

In the present study, the plasma concentrations of corticosterone were increased by restraint (Table 2, Figure 1). Across vertebrates, the hypothalamo–pituitary–adrenal (HPA) axis involves the corticotropin-releasing hormone, the adrenocorticotropic hormone and a glucocorticoid (cortisol, corticosterone or 11-deoxycortisol, depending on the species), with stressors stimulating the axis (wild birds, e.g., sparrows [39]; Reptilia and Amphibia [40]; Osteichthyes (boney fish), e.g., trout [41], reviewed in [42]; Chondrichthyes (cartilaginous fish) [43]; Agnatha (jawless fish) [44]). In a similar manner, for instance, the plasma concentrations of corticosterone in chickens are elevated by both acute and chronic immobilization stress [4,5,6].

The plasma concentrations of Met-enkephalin together with PENK expression were increased in stressed chickens (Table 2, Figure 1) as observed previously [37]. There is evidence of a crosstalk between the opioid/Met-enkephalin system and the HPA axis across the vertebrates. For instance, challenging immature trout with the opioid morphine or the met-enkephalin analogue D-Ala2, Met5-enkephalinamide was followed by increases in the circulating glucocorticoid concentrations [45]. Moreover, morphine stimulates the release of CRH from the hypothalamus in vitro (trout [46]; rat [47]) and an increase in the circulating concentrations of corticosterone (rat [48]). It is not clear whether the increases in the plasma concentrations of Met-enkephalin in stressed chickens (Table 2, Figure 1) are secondary to those in the plasma concentrations of corticosterone, insulin or glucose.

There were relationships between the plasma concentrations of Met-enkephalin together with PENK expression and those of either insulin or glucose. There is other evidence that Met-enkephalin influences the carbohydrate metabolism and its control by insulin. Met-enkephalin increased the plasma concentrations of glucose in rats [49] and glucose release from rat hepatocytes in vitro [50]. The plasma concentrations of Met-enkephalin are increased in insulin-dependent diabetics and further elevated following a meal and insulin therapy, but depressed in neuropathic patients [51,52]. Moreover, the plasma concentrations of glucose following a glucose load or a meal was reduced by a Met-enkephalin analogue [53].

There were also strong relationships between the plasma concentrations of insulin and those of glucose either 30 or 40 min after initiation of the treatments but not prior to treatment. Similarly, relationships between the circulating concentrations of insulin and those of gastric inhibitory peptide (GIP) or glucagon-like peptide (GLP) were reported in chickens [54], as well as between the circulating concentrations of insulin and those of uric acid [55].

It is hypothesized that the stress-induced increase in the plasma concentrations of glucose is secondary either to the release of epinephrine/norepinephrine or to the production of glucocorticoids. The former possibility is unlikely, as chickens challenged with epinephrine failed to exhibit changes in the circulating concentrations of glucose [56]. Corticosterone is considered the main physiological regulator in response to stress, rather than glucose, based on studies in sparrows [39]. Moreover, stress influence on insulin secretion in rats is at least partially mediated by glucocorticoids [57]. Moreover, stress influences glucose-stimulated insulin release in rats [58]. The increase in the plasma concentrations of insulin in stressed chickens (Table 2, Figure 2) were likely secondary to corticosterone-induced hyperglycemia. 

The administration of naltrexone, blocking the effects of Met-enkephalin, was followed by increases in the plasma concentrations of corticosterone but attenuated the increase in the plasma concentrations of corticosterone in restrained chickens (Table 2, Figure 1). There was no increase in the plasma concentrations of Met-enkephalin following naltrexone administration per se, but naltrexone decreased the magnitude of the increase in the plasma concentrations of Met-enkephalin in restrained chickens (Table 2, Figure 2). Naltrexone had no effect on the plasma concentrations of insulin (Table 2, Figure 3). There was an increase in the plasma concentrations of glucose in male but not in female chickens following naltrexone administration (Table 2, Figure 3). Naltrexone again attenuated the increase in the plasma concentrations of glucose (Table 2).

The plasma concentrations of insulin were increased in chickens subjected to restraint (Table 2 Figure 3) and, to a lesser extent, repeated blood sampling (Table 2, Figure 2). The basis for these effects is not totally clear. They could be secondary to the increase in the plasma concentrations of glucose. Despite the absence of GLUT 4 in chickens [59,60], insulin secretion is increased by either glucose or the glycolytic hormone glucagon in chickens [61,62]. Exogenous glucocorticoids were reported to induce insulin release in chickens, with circulating concentrations of insulin being elevated following the administration of corticosterone [31,32,34] or dexamethasone [33,35]. In mammals, exogenous glucocorticoids were reported to increase or to decrease insulin secretion. Examples of studies in which glucocorticoids increased insulin release include the following. The administration of dexamethasone in vivo in human subjects for three days was accompanied by increases in the plasma concentrations of insulin [63]. Moreover, there was marked hyperglycemia in men receiving prednisolone administration for 2 or 15 days [64]. Similarly, dexamethasone administration was followed by elevated circulating concentrations of insulin in humans [65] and rats [66]. The in vitro release of insulin was depressed in adrenalectomized rats, and this was overcome by cortisol replacement therapy in vivo [67]. In contrast, dexamethasone depressed insulin release from mouse islets in vitro at both physiological and high concentrations of glucose [68]. Corticosterone administration depressed glucose-stimulated insulin release in mice [69]. Moreover, there was both a larger decrease in insulin secretion and a greater expression of a reporter gene in the β cells in mice over-expressing the glucocorticoid receptor and challenged with the glucocorticoid dexamethasone [70]. Overall, the ability of glucocorticoids to increase the circulating concentrations of insulin is consistent with the increase in the plasma concentrations of insulin being secondary to the increase in the plasma concentrations of corticosterone (Table 2, Figure 3).

The plasma concentrations of glucose were increased in chickens subjected to restraint and also elevated with repeated blood sampling. This was observed despite the plasma concentrations of insulin being increased (Table 2). This is consistent with stress inducing insulin resistance. In chickens, as in mammals, the plasma concentrations of glucose are depressed following insulin administration (chickens [55,71]. There is other evidence that glucocorticoids induce insulin resistance in chickens. The administration of the exogenous glucocorticoid dexamethasone increased the plasma concentrations of glucose [33,35]. Moreover, dexamethasone decreased the expression of GLU1 in the presence of insulin in myoblasts in vitro [34]. Similarly, stresses are associated with insulin resistance in mammals. For instance, insulin resistance was observed in mice subjected to inescapable foot shock stress, a model for acute psychological stress [55]. Moreover, both insulin resistance and increased endogenous glucocorticoid concentrations were observed in a mouse model of chronic psychological stress consisting of repeated acoustic and restraint stresses [72].

It is argued that the effects of restraint stress on the plasma concentrations of corticosterone and Met-enkephalin (Table 2) are quickly reversed once the stressful situation is terminated. The decline in the plasma concentrations of corticosterone and Met-enkephalin are presumed to reflect the clearance of these hormones. The observed decrease in the plasma concentrations of corticosterone in female chickens is consistent with a half-life of ~10 min (Table 2). This is similar to the half-life of 6 min reported in the Pekin duck [73] but shorter than that of 22 min reported in broiler chickens [74]. However, despite the marked decreases in the plasma concentrations of glucose (Table 2), there was either a smaller decrease (females) or an increase (males) in the plasma concentrations of insulin during the recovery period post the restraint stress (Table 2). This may reflect a time lag in either the hyperglycemia or the corticosterone stimulation of insulin secretion. 

## 5. Conclusions

The present study demonstrated acute effects of restraint stress on the plasma concentrations of Met-enkephalin and corticosterone in growing chickens. The plasma concentrations of insulin were also increased by stressors, namely, restraint stress and repeated blood sampling, in chickens. The present results provide further support for a crosstalk between insulin, Met-enkephalin and the HPA axis. Furthermore, it is suggested that plasma insulin concentration may be a useful parameter indicating stress and, consequently, welfare in chickens. 

## Figures and Tables

**Figure 1 animals-14-00752-f001:**
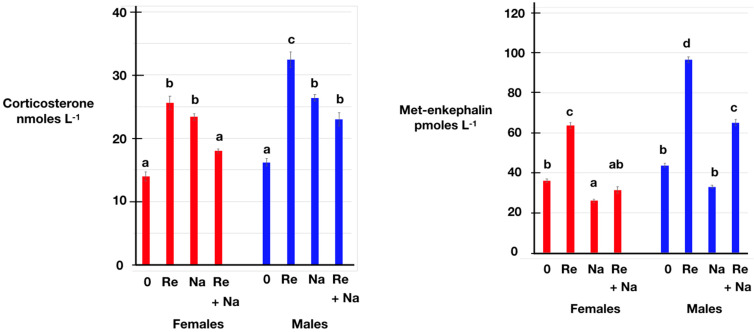
Effect of restraint stress for 30 min in the presence or absence of naltrexone on the plasma concentrations of corticosterone (**left**) and Met-enkephalin (**right**) in 14-week-old female and male chickens. [key: 0 control, Re—restraint stress, Na—naltrexone (2 mg/kg b.w. i.v.)]. Vertical bars indicate SEM (*n* = 5). Different letters indicate a significant difference, *p* < 0.05.

**Figure 2 animals-14-00752-f002:**
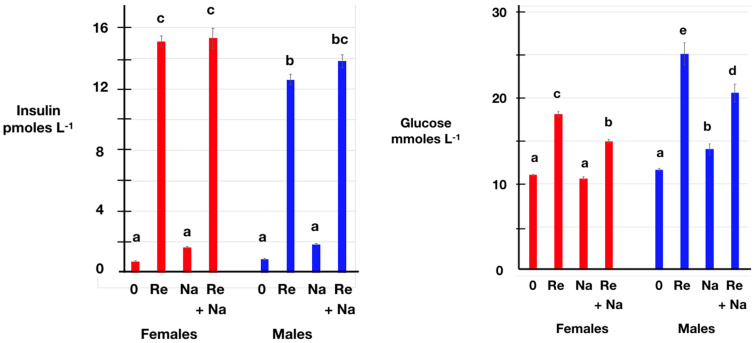
Effect of restraint stress for 30 min in the presence or absence of naltrexone on the plasma concentrations of insulin (**left**) and glucose (**right**) in 14-week-old female and male chickens. [key: 0 control, Re—restraint stress, Na—naltrexone (2 mg/kg, b.w. i.v.)]. Vertical bars indicate SEM (*n* = 5). a, b, c, d, e Different superscript letters indicate a significant difference, *p* < 0.05.

**Figure 3 animals-14-00752-f003:**
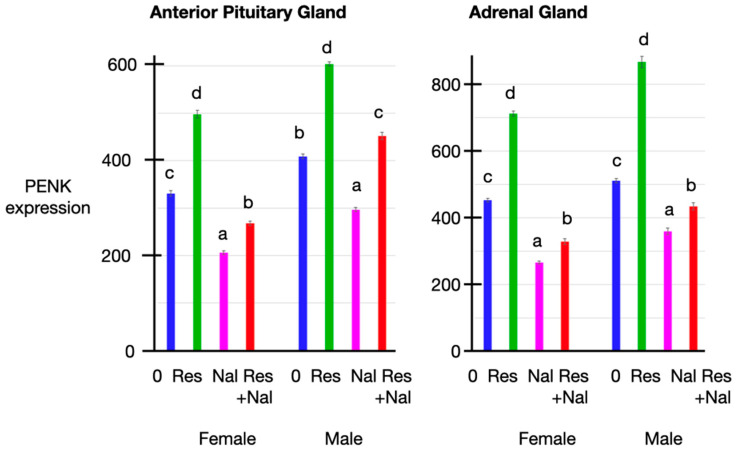
Effect of restraint and/or naltrexone administration (*n* = 3 per sex per treatment) on PENK expression (PSL/mm^2^) in anterior pituitary and adrenal tissue in chickens. [a, b, c, d—Different superscript letter indicates differences, *p* < 0.05; Res—chickens subjected to restraint stress, Nal—chickens receiving naltrexone; vertical bars indicate SEM (*n* = 3 per sex per treatment)].

**Table 1 animals-14-00752-t001:** Basal plasma concentrations of corticosterone, Met-enkephalin, glucose and insulin in female (*n* = 20) and male (*n* = 20) chickens (mean ± SEM).

	Female	Male
Corticosterone nmoles L^−1^	15.0 ± 0.41	15.0 ± 0.34
Met-enkephalin pmoles L^−1^	37.1 ± 0.58 ^a^	41.9 ± 0.73 ^b^
Insulin pmoles L^−1^	0.696 ± 0.019	0.714 ± 0.023
Glucose mmoles L^−1^	9.89 ± 0.094 ^a^	11.2 ± 0.109 ^b^

^a,b^ Different superscript letters indicate a significant sex difference, *p* < 0.0001.

**Table 2 animals-14-00752-t002:** Changes in the plasma concentrations of corticosterone, Met-enkephalin, insulin and glucose in 14-week-old chickens following multiple blood sampling (controls), restraint stress and/or naltrexone administration ^#^ [mean ± (*n* = 5) SEM].

Treatments	Pre-Treatment	Treatment 30 min	10 min Recovery	Pre-Treatment	Treatment 30 min	10 min Recovery
	Plasma concentrations of corticosterone nmoles L^−1^			Plasma concentrations of Met-enkephalin pmoles L^−1^		
Controls						
Female	15.0 ± 0.71	14.0 ± 0.71 ^p^	13.4 ± 0.93	38.6 ± 0.81	36.0 ± 1.00 ^p^	35.0 ± 1.14
Male	13.6 ± 0.51	16.2 ± 0.58 ^p^	15.0 ± 0.71	41.6 ± 0.81	43.8 ± 1.07 ^pq^	44.8 ± 1.07
Restraint						
Female	14.0 ± 0.71 ^a^	25.6 ± 1.03 ^bq^	13.0 ± 0.71 ^a^	37.0 ± 1.22 ^a^	63.6 ± 1.44 ^br^	46.2 ± 1.56 ^a^
Male	15.8 ± 0.86 ^a^	32.4 ± 1.21 ^cr^	23.5 ± 1.64 ^b^	37.8 ± 1.07 ^a^	96.4 ± 1.44 ^cs^	88.0 ± 1.14 ^b^
Naltrexone						
Female	16.8 ± 0.61 ^b^	23.4 ± 0.51 ^cpq^	12.6 ± 0.51 ^a^	35.6 ± 1.36 ^b^	26.0 ± 0.71 ^ap^	31.6 ± 0.51 ^b^
Male	15.4 ± 0.51 ^a^	26.4 ± 0.51 ^bq^	17.8 ± 0.37 ^a^	43.2 ± 0.86 ^b^	33.0 ± 0.71 ^ap^	41.6 ± 1.50 ^b^
Restraint + naltrexone						
Female	14.1 ± 0.71	18.0 ± 0.32 ^pq^	16.8 ± 0.37	37.4 ± 1.12 ^b^	31.4 ± 1.57 ^ap^	27.8 ± 0.86 ^a^
Male	15.4 ± 0.71	23.0 ± 0.71 ^pq^	18.6 ± 0.93 ^pq^	45.0 ± 0.71 ^a^	65.0 ± 1.58 ^cr^	59.2 ± 1.24 ^b^
Effects		3-way ANOVA F = (*P*</=)			3-way ANOVA F = (*P*</=)	
Sex		56.9 (<0.001)			540 (<0.001)	
Restraint		71.1 (<0.001)			1132 (<0.001)	
Naltrexone		1.33 (0.257)			583 (<0.001)	
Sex × restraint interaction		8.57 (<0.01)			218 (<0.001)	
Sex × naltrexone interaction		0.20 (=0.66)			0	
Restraint × naltrexone interaction		263 (<0.001)			150 (<0.001)	
Sex × restraint interaction × naltrexone interaction		1.33 (=0.26)			0.21 (=0.65)	
	Plasma concentrations of insulin pmoles L^−1^			Plasma concentrations of glucose mmoles L^−1^		
Controls						
Female	0.68 ± 0.031 ^a^	0.74 ± 0.03 ^bp^	0.83 ± 0.03 ^b^	10.3 ± 0.10 ^a^	11.1 ± 0.06 ^bp^	11.6 ± 0.13 ^c^
Male	0.71 ± 0.052 ^a^	0.89 ± 0.05 ^ap^	1.22 ± 0.06 ^b^	11.5 ± 0.06 ^a^	11.7 ± 0.12 ^ap^	12.2 ± 0.13 ^b^
Restraint						
Female	0.73 ± 0.040 ^a^	15.1 ± 0.35 ^bq^	14.0 ± 1.06 ^b^	9.7 ± 0.13 ^a^	18.1 ± 0.28 ^cr^	13.8 ± 1.16 ^b^
Male	0.68 ± 0.056 ^a^	12.6 ± 0.36 ^bq^	15.4 ± 1.90 ^b^	11.4 ± 0.07 ^a^	25.1 ± 0.20 ^cs^	13.8 ± 0.55 ^b^
Naltrexone						
Female	0.71 ± 0.035 ^a^	1.66 ± 0.06 ^bp^	1.85 ± 0.08 ^c^	9.4 ± 0.06 ^a^	10.6 ± 0.17 ^bp^	11.7 ± 0.12 ^c^
Male	0.76 ± 0.039 ^a^	1.85 ± 0.04 ^bp^	2.38 ± 0.04 ^c^	11.0 ± 0.14 ^a^	20.6 ± 0.14 ^cr^	14.0 ± 0.14 ^b^
Restraint + naltrexone						
Female	0.66 ± 0.022 ^a^	15.3 ± 0.67 ^q^	18.8 ± 0.67 ^c^	10.2 ± 0.14 ^a^	15.0 ± 0.17 ^bq^	14.0 ± 0.35 ^b^
Male	0.72 ± 0.042 ^a^	13.8 ± 0.41 ^bq^	17.6 ± 0.35 ^c^	11.0 ± 0.14 ^a^	20.6 ± 0.11 ^cq^	14.0 ± 0.35 ^b^
Effects		3-way ANOVA F= (*P*</=)			3-way ANOVA F= (*P*</=)	
Sex		14.0 (=0.001)			14.5 (=0.001)	
Restraint		3022 (<0.001)			235 (<0.001)	
Naltrexone		12.0 (<0.01)			15.1 (<0.001)	
Sex × restraint interaction		20.1 (<0.001)			8.6 (< 0.01)	
Sex × naltrexone interaction		1.34 (=0.25)			0.80 (=0.38)	
Restraint × naltrexone interaction		0.30 (=0.59)			F = 2.06 (=0.16)	
Sex × restraint interaction × naltrexone interaction		F = 1.07 (=0.31)			F = 7.4 (<0.05)	

^#^ Sampled during pre-treatment, after 30 min and 40 min. ^a,b,c^ Different superscript letters indicate differences with time; ^p,q,r,s^ different superscript letters indicate a significant difference.

## Data Availability

The data presented in this study are available on request from the corresponding authors.

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
