# Peer review of "Effects of Restraint Stress on Circulating Corticosterone and Met Enkephalin in Chickens: Induction of Shifts in Insulin Secretion and Carbohydrate Metabolism"

_animals, 2024, doi:10.3390/ani14050752_

Round 1
Reviewer 1 Report
Comments and Suggestions for Authors
In this study, the authors studied the effects of restraint stress on animal metabolism and insulin secretion, which is helpful for coping with stress in farming, and also helps to prevent harm caused by stress, which is of great significance for the development of animal welfare.
I have only a few minor suggestions for the authors to consider:
1. Line 40 read "depression anxiety-like behavior" with "anxiety-like behavior".
2. Line 43: Remove "and".
3. Line 47: Please specify whether an increase or decrease in "Escherichia coli lipopolysaccharide" affects its concentration.
4. Line 80: The unit of Celsius is written incorrectly, please correct it.
5. Line 91: What does the phrase "1;2;3" mean after the colon.
6. Line 93: "3.000" to "3000".
7. hange the line 118 "P" to italic, and check the full text.
8. Please align the data in Table 3.
9. The citation for line 408 is not found in Pub.
Author Response
We thank the reviewer for their helpful comments.
- Line 40 read "depression anxiety-like behavior" with "anxiety-like behavior".
Response: The sentence has been clarified to meet the reviewer’s concern. In particular, the word “depressed” has been replaced by “reduced”.
- Line 43: Remove "and".
Response: The word “and” has been removed.
- Line 47: Please specify whether an increase or decrease in "Escherichia coli lipopolysaccharide" affects its concentration.
Response: The sentence has been clarified adding “challenged with” added prior to Escherichia coli lipopolysaccharide
- Line 80: The unit of Celsius is written incorrectly, please correct it.
Response: 20ºC has been changed to 20o C.
- Line 91: What does the phrase "1;2;3" mean after the colon.
Response: 1, 2, and 3 have been removed.
- Line 93: "3.000" to "3000".
Response: Corrected as requested.
- Change the line 118 "P" to italic, and check the full text.
Response: Corrected as requested.
- Please align the data in Table 3.
Response: Data are aligned in all tables now.
- The citation for line 408 is not found in Pub. (Reference 37)
Response: A citation is now included on line 97.
Reviewer 2 Report
Comments and Suggestions for Authors
I only have minor comments on this manuscript. I think the discussion could be improved by discussing how non-mammalian species (other birds, reptiles, fish) respond to similar stressors of confinement. I am not how much research is out there about these "non-traditional" species, but a more comparative angle would make this more interesting.
line 43 insert axis
48 "chickens.....in turkeys" chickens housed together with turkey's?
57 delete second also
65 what is the difference between pharmacological effects and physiological effects?
77-79 It is not clear to me whether the birds were caged individually housed prior to the treatment. This part is not very clear to me.
128 Whate about met enkephalin and insulin?
Table 3 incorrect labeling - corticosterone is labeled 3 times should be insulin and glucose?
161 they were reduced
179 delete and
194 samplings
241 add between after relationships
257 lesser extent
Author Response
We thank the reviewer for their helpful comments.
I only have minor comments on this manuscript. I think the discussion could be improved by discussing how non-mammalian species (other birds, reptiles, fish) respond to similar stressors of confinement. I am not how much research is out there about these "non-traditional" species, but a more comparative angle would make this more interesting.
Response: A comparative approach has been added to the discussion with over 15 additional references and three new paragraphs.
line 43 insert axis
Response: Corrected as suggested.
48 "chickens.....in turkeys" chickens housed together with turkey's?
Response:
This is revised making clear that it is in separate studies.
57 delete second also
Response: Corrected as suggested.
65 what is the difference between pharmacological effects and physiological effects?
Response: Clarified with definition of pharmacological added.
77-79 It is not clear to me whether the birds were caged individually housed prior to the treatment. This part is not very clear to me.
Response: The word “separately” was changed to the more definitive term – “individually”.
128 What about met enkephalin and insulin?
Response: Clarified with all relationships compared.
Table 3 incorrect labeling - corticosterone is labeled 3 times should be insulin and glucose?
Response: We are very gratefully for the reviewer find this error. It is now corrected.
161 they were reduced
Response: Corrected.
179 delete and
Response: Corrected.
194 samplings
Response: We respectfully disagree with the English usage.
241 add between after relationships
Response: Corrected.
257 lesser extent
Response: Corrected.
Reviewer 3 Report
Comments and Suggestions for Authors
In general, the manuscript entitled ‘Effects of Restraint Stress on Circulating Corticosterone and Met Enkephalin in Chickens: Induction of Shifts in Insulin Secretion and Carbohydrate Metabolism’ presents an interesting and valuable topic, which may be of interest to readers. However, the reviewed manuscript requires solid corrections in relation to the methodological and statistical layer, because without a detailed description of the methodological part, it is difficult to evaluate the obtained results (unclear experimental setup, lack of interaction).
Abstract:
There is no information on the number of chickens in the experimental design per groups and the number of replications. This should be added.
Introduction:
L38-39: Please provide some examples so that the reader does not have to verify the cited sources.
Materiał i metody:
L76-84: The experimental design is not fully understandable and requires clarification. The authors reported that 20 male and 20 female chickens were used (n = 40); however, after the adaptation period 4 treatment groups were created with 5 individuals each. This makes the impression that 20 birds participated in the experiment. If this was the case, with n = 5 birds per group, the number of females and males in the one group is not equal, which may affect the result obtained (n = 5 is also a slightly small number). Therefore, it is necessary to provide: how many individuals each group had, because in the results section the authors provided males (n = 20) and females (n = 20).
L108-L115: Indicate the experimental factors. Since a three-way ANOVA was used, it is necessary to specify which factors were responsible for the effect obtained + what interactions occurred. Here it looks like a 1-way ANOVA (4 treatment groups), perhaps an additional factor is sex or repeatability of measurements (which, with the option of repeated measurements, is not selected as an experimental factor), as can be assumed based in Table 3? Or maybe a total of 8 groups were created, each treatment separately for males and females (then n = 40)? Please clarify this issue in the methodology.
Indicate whether the data were checked for normal distribution? If so, please provide this information in this section. Provide the software in which the data evaluation was performed.
Results:
Table 3: At this point, the overall experiment is a bit more understandable: it seems that the authors were testing the effect of treatment, gender, and repeated measures (actually two experimental factors but evaluated for repeated measures). If so, the whole thing should be described in the methodology. Additionally, the table should include the exact P-values for the experimental factors and interactions. Therefore, provide the exact P values for each experimental factors and their interaction in Tables where ANOVA was used, then the reader knows what factor influences the result. Moreover, the letter designations should not show differences between groups/treatments (1-way ANOVA result, must be removed from Table 3), but between the main interactions (without including repeated measurements). Differences in the effects of experimental factors should be shown only by P values in such case.
In other words: mean values for groups/treatments, letters for interactions (and repeated measurements, if applicable), and P values for the experimental factors. This clearly shows if the each factor works or does not work, which fundamentally facilitates the interpretation of the obtained results.
Other minor issues:
The manuscript contains many editorial errors. Please to check the text carefully, as numerous spaces (double spaces? and typos) are present in the text of the manuscript.
Examples:
L21, 24, 25, 26, 35, 38, 48, 51, 53, 55, 60, 88, 98, 145, 183, 201, and more.
Author Response
Reviewer 3
In general, the manuscript entitled ‘Effects of Restraint Stress on Circulating Corticosterone and Met Enkephalin in Chickens: Induction of Shifts in Insulin Secretion and Carbohydrate Metabolism’ presents an interesting and valuable topic, which may be of interest to readers. However, the reviewed manuscript requires solid corrections in relation to the methodological and statistical layer, because without a detailed description of the methodological part, it is difficult to evaluate the obtained results (unclear experimental setup, lack of interaction).
Response: We thank the reviewer for their helpful comments.
Abstract:
There is no information on the number of chickens in the experimental design per groups and the number of replications. This should be added.
Response: This is now added as is the experimental design.
Introduction:
L38-39: Please provide some examples so that the reader does not have to verify the cited sources.
Response: The following statement has been added to the narrative:
Anxiety related behaviors in mice can be assessed by avoidance of heights and open spaces and preference for dark enclosed spaced in an elevated plus maze [13, 14] together with hyperactivity activity, in for instance, in light/dark transition tests and open field tests [14, 15]. In mice, depression related behaviors include the following: Porsolt forced swim test, immobility in the tail suspension test and sucrose preference test [14, 15].
Materials and Methods
L76-84: The experimental design is not fully understandable and requires clarification. The authors reported that 20 male and 20 female chickens were used (n = 40); however, after the adaptation period 4 treatment groups were created with 5 individuals each. This makes the impression that 20 birds participated in the experiment. If this was the case, with n = 5 birds per group, the number of females and males in the one group is not equal, which may affect the result obtained (n = 5 is also a slightly small number). Therefore, it is necessary to provide: how many individuals each group had, because in the results section the authors provided males (n = 20) and females (n = 20).
Response: We have clarified the materials and methods section in an attempt make clear that there are five male and five females per treatment.
L108-L115: Indicate the experimental factors. Since a three-way ANOVA was used, it is necessary to specify which factors were responsible for the effect obtained + what interactions occurred. Here it looks like a 1-way ANOVA (4 treatment groups), perhaps an additional factor is sex or repeatability of measurements (which, with the option of repeated measurements, is not selected as an experimental factor), as can be assumed based in Table 3? Or maybe a total of 8 groups were created, each treatment separately for males and females (then n = 40)? Please clarify this issue in the methodology.
Response: We have clarified the materials and methods section in an attempt make clear that there are five male and five females per treatment (total n = 40). Moreover, the 3-way ANOVAs are shown.
Indicate whether the data were checked for normal distribution? If so, please provide this information in this section.
Response: This is now included.
Provide the software in which the data evaluation was performed.
Response: The software is now included.
Results:
Table 3: At this point, the overall experiment is a bit more understandable: it seems that the authors were testing the effect of treatment, gender, and repeated measures (actually two experimental factors but evaluated for repeated measures). If so, the whole thing should be described in the methodology. Additionally, the table should include the exact P-values for the experimental factors and interactions. Therefore, provide the exact P values for each experimental factors and their interaction in Tables where ANOVA was used, then the reader knows what factor influences the result. Moreover, the letter designations should not show differences between groups/treatments (1-way ANOVA result, must be removed from Table 3), but between the main interactions (without including repeated measurements). Differences in the effects of experimental factors should be shown only by P values in such case.
In other words: mean values for groups/treatments, letters for interactions (and repeated measurements, if applicable), and P values for the experimental factors. This clearly shows if the each factor works or does not work, which fundamentally facilitates the interpretation of the obtained results.
Response: The 3-way ANOVAs are now shown in Table 2. A superscript letters system, (p, q, r and s) is employed to show differences between treatment and sexes are shown separated by Tukey’s HSD.
Other minor issues:
The manuscript contains many editorial errors. Please to check the text carefully, as numerous spaces (double spaces? and typos) are present in the text of the manuscript.
Examples:
L21, 24, 25, 26, 35, 38, 48, 51, 53, 55, 60, 88, 98, 145, 183, 201, and more.
Response: The double spaces between sentences have been removed.
Reviewer 4 Report
Comments and Suggestions for Authors
I have to say that I was surprised at the lack of originality with this study. I believe that these findings might have been new (but still not particularly exciting) in the 1980s comparative physiology literature. The authors need to do a better job of convincing me that this paper adds to that literature.
There needs to be more detail in the M&Ms about how the control and Naltrexone only chickens were bloodsampled as it seems impossible not to have to physically restrain the birds to do so. The fact that they might have been in a 60 x 60 x 60 cage with visual and auditory cues available compared to being in a 30 x 30 x 30 dark and silent box doesn't seem to be without stress. Were the birds cannulated during the pre-treatment sampling event to reduce handling stress?
Unfortunately, the disappointments continue. Table 1 shows differences in levels of the 4 parameters between male and female chickens in the basal/pre-treatment samples but there is no analysis of the differences in those parameters within each sex between treatment groups and, to me, this is critical to establishing whether it is appropriate to pool them for an overall average. Further, there is no indication in Table 3 that each average and SEM is of 5 birds.
The error in Table 3 with "Plasma concentrations of corticosterone nmoles L-1" in the heading for insulin and glucose data is an annoying demonstration of poor proofreading.
The more annoying aspect of this manuscript is that the "10 min recovery" data are almost completely ignored. If you have 3 serial samples from the same birds you can (and should) show them as joined datapoints on a line graph to better demonstrate if there are differences in "area under the curve" for each sex and treatment comparison. This would be a much more appropriate way to present the data. There is no need for repeating information in Tables and Figures, and the correlational analysis adds nothing to our understanding of what is going on.
The conclusion that naltrexone induces insulin resistance is completely speculative. This suggestion should be de-emphasized.
Author Response
I have to say that I was surprised at the lack of originality with this study. I believe that these findings might have been new (but still not particularly exciting) in the 1980s comparative physiology literature. The authors need to do a better job of convincing me that this paper adds to that literature.
Response: We were disappointed by the tone of the review We have, however, taken the reviewer’s comments to heart. We have tightened the text and removed some of the regressions including the tables. We have also added data on the expression of the proenkephalin gene (PENK) in the anterior pituitary gland and adrenal gland. This is not only novel but also exciting.
We have revised the manuscript in an attempt to address the reviewer’s comments and to improve the clarity of presentation. The findings are new and have not been reported by us or others:
- Stress increases both plasma concentrations of insulin in chickens.
- Stress increases both plasma concentrations of glucose in chickens.
- Stress increases both plasma concentrations of Met-enkephalin in chickens.
- There are relationships between plasma concentrations of corticosterone and those of Met-enkephalin, insulin and glucose together with expression of PENK.
The old tables 2, 4 and 5 has been removed as has the narrative about it.
There needs to be more detail in the M&Ms about how the control and Naltrexone only chickens were blood sampled as it seems impossible not to have to physically restrain the birds to do so.
Response: We have clarified that the birds were transitorily manually held. Moreover, we report shifts in circulating concentrations of the three hormones together with those of glucose which are consistent with mild stress (new table 3).
Response: We have clarified that the birds were transitorily manually held. Moreover, we report shifts in circulating concentrations of the three hormones together with those of glucose which are consistent with mild stress (new table 3).
The fact that they might have been in a 60 x 60 x 60 cage with visual and auditory cues available compared to being in a 30 x 30 x 30 dark and silent box doesn't seem to be without stress. Were the birds cannulated during the pre-treatment sampling event to reduce handling stress?
Response: Birds were not cannulated.
Unfortunately, the disappointments continue. Table 1 shows differences in levels of the 4 parameters between male and female chickens in the basal/pre-treatment samples but there is no analysis of the differences in those parameters within each sex between treatment groups and, to me, this is critical to establishing whether it is appropriate to pool them for an overall average.
Response: We respectfully disagree. Given that the pre-sampling was from birds that were yet to be treated, pooling seems reasonable.
Further, there is no indication in Table 3 that each average and SEM is of 5 birds.
The error in Table 3 with "Plasma concentrations of corticosterone nmoles L-1" in the heading for insulin and glucose data is an annoying demonstration of poor proofreading.
Response: We apologize to the reviewer for the mistake. This is now corrected in new table 2.
The more annoying aspect of this manuscript is that the "10 min recovery" data are almost completely ignored. If you have 3 serial samples from the same birds you can (and should) show them as joined datapoints on a line graph to better demonstrate if there are differences in "area under the curve" for each sex and treatment comparison. This would be a much more appropriate way to present the data.
Response: We respectfully disagree with the reviewer. We have only one data point during the restraint stress, namely the 30-minute time point. We, therefore, focused statistical analysis to the 30-minutes time point.
There is no need for repeating information in Tables and Figures, and the correlational analysis adds nothing to our understanding of what is going on.
Response: We respectfully disagree. The figures are present to make the paper more reader friendly.
The conclusion that naltrexone induces insulin resistance is completely speculative. This suggestion should be de-emphasized.
Response: We have deemphasized the entire discussion on insulin resistance removing six references and much of one paragraph.
- Iowa is in the bottom quintile for percentage of the population with a college degree (According to USA Facts)
- Iowa is in the bottom quintile for number of doctors per capita (According to Association of American Medical Colleges State Physician Workforce Data Report).
Round 2
Reviewer 3 Report
Comments and Suggestions for Authors
Dear Authors,
Overall, the manuscript has been mostly satisfactorily revised, so I have no further comments.
Please revise the manuscript for editorial purposes, as there are minor errors.
E.g.: L276-277, where the wording ‘the increases’ is repeated.
L101: ‘(30 x 30 x30 cm)’ -> ‘(30 x 30 x 30 cm)’
L250: ‘[adjusted R2 = 0.867 (P < 0.001])’ -> ‘[adjusted R2 = 0.867 (P < 0.001)]’.
Author Response
Response to reviewer 3.
We thank the reviewer for their helpful comments.
E.g.: L276-277, where the wording ‘the increases’ is repeated.
Response: Corrected
L101: ‘(30 x 30 x30 cm)’ -> ‘(30 x 30 x 30 cm)’
Response: Corrected
L250: ‘[adjusted R2 = 0.867 (P < 0.001])’ -> ‘[adjusted R2 = 0.867 (P < 0.001)]’.
Response: Corrected
Reviewer 4 Report
Comments and Suggestions for Authors
The pENK data is novel and improves the manuscript. It makes me wonder why wasn't it included in the original version?
The resistance to accept several of my suggestions to improve the manuscript is noted. The citation of many comparative physiology/endocrinology papers of stress responses of various vertebrates to acute and chronic stressors from the 1980s and 1990s that support some of the points made in the Discussion just reinforce my suggestion that this study has many parallels to that era. And as a product of that era myself, I can say that they were great papers. But hasn't the field moved on in the past 30 years?
The unwillingness to consider the 10 mins post treatment data in any meaningful way makes it redundant and it should be excluded.
The inclusion of Tables and Figures showing the same information is becoming more commonplace but it is not a trend I support.
Author Response
Response to reviewer 4.
We thank the reviewer for their thoughtful comments.
The PENK data is novel and improves the manuscript. It makes me wonder why wasn't it included in the original version?
Response: We appreciate the reviewer noting that “PENK data is novel and improves the manuscript”. The comments of the reviewer served as encouragement to add the PENK expression data. The PENK expression data was not included initially as the manuscript as it focused on plasma concentrations of hormones and glucose.
The resistance to accept several of my suggestions to improve the manuscript is noted.
Response: We apologize if the reviewer feels we have not accepted or responded to their suggestions.
The citation of many comparative physiology/endocrinology papers of stress responses of various vertebrates to acute and chronic stressors from the 1980s and 1990s that support some of the points made in the Discussion just reinforce my suggestion that this study has many parallels to that era. And as a product of that era myself, I can say that they were great papers. But hasn't the field moved on in the past 30 years?
Response:
- Discussion of “many comparative physiology/endocrinology papers of stress responses of various vertebrates to acute and chronic stressors” together with appropriate citations was added to meet the recommendation of reviewer 2.
- We agree that the field, particularly in rodent models, has moved forward and we included discussion of this in the introduction.
- It is difficult to address the reviewer’s inference that the work belongs to the 1980s and 1990s. The restraint induced shifts in plasma concentrations of Met-enkephalin, insulin and glucose are novel and also relevant to the welfare of poultry. Indeed, this is the first definitive of a stressor inducing hyperglycemia in chickens.
The unwillingness to consider the 10 mins post treatment data in any meaningful way makes it redundant and it should be excluded.
Response: We apologize if the reviewer feels we have not adequately covered the 10 min post treatment but have accepted their suggestions. We have attempted to address the spirit of their concern adding to the discussion lines 250 - 260.
The inclusion of Tables and Figures showing the same information is becoming more commonplace but it is not a trend I support.
Response: As this is a commonplace situation, we find difficult to address the reviewer’s point.